# Food and Beverage Consumption Habits through the Perception of Health Belief Model (Grab Food or Go Food) in Surabaya and Pasuruan

**DOI:** 10.3390/nu14214482

**Published:** 2022-10-25

**Authors:** Trias Mahmudiono, Qonita Rachmah, Diah Indriani, Erwanda Anugrah Permatasari, Nur Alifia Hera, Hsiu-Ling Chen

**Affiliations:** 1Department of Nutrition, Faculty of Public Health, Universitas Airlangga, Surabaya 63138, Indonesia; 2Department of Epidemiology, Biostatistics, Health Promotion and Behavioral Sciences, Faculty of Public Health, Universitas Airlangga, Surabaya 63138, Indonesia; 3Department of Safety/Hygiane and Risk Management, National Cheng Kung University, Tainan City 701, Taiwan

**Keywords:** health belief model, consumption habits, nutritional status, health and wellbeing

## Abstract

Background: The metaverse as a digital environment for Industrial Revolution 4.0 is one major form of use of the internet. There are 202.6 million internet users in Indonesia in 2021, or 73.7% of the total population. A total of 138.1 million Indonesians aged 18–64 years have used the internet to make purchases through e-commerce and 74.4% make food purchases through online food delivery applications. Most of the foods sold in online applications are foods that are high in carbohydrate and fat, but with fewer vegetables and fruits. So, it can be concluded that the food sold is energy dense, nutrition poor. Because of that, people run the risk of degenerative diseases such as hypertension, diabetes mellitus, stroke, and others. By using the health belief model approach, this study aims to analyze the association between the habits of consuming food and beverages purchased online through the Grab Food or Go Food applications with the nutritional status of the people in Surabaya and Pasuruan, East Java, Indonesia. Methods: This research is quantitative research using a cross-sectional approach. Data collection was carried out offline using paper questionnaires and analysis with SPSS. Result: There was significant association between the characteristics of the respondents (age, marriage status, profession, education level, income, and allowance) and health beliefs. However, there was no association between health beliefs and the frequency of online orders. Finally, no significant association was found between perceived susceptibility, perceived severity, perceived benefit, perceived barrier, cues to action, self-efficacy and nutritional status. Thus, it is still important for the government to increase socialization and education in the importance of balanced nutrition and nutritional status so that people can protect themselves and prevent the onset of degenerative diseases.

## 1. Introduction

The metaverse as a digital environment for Industrial Revolution 4.0 is one major form of use of the internet. There are 202.6 million internet users in Indonesia in 2021, or 73.7% of the total population [1]. The development of the internet supports the development of people’s lives and triggers a shift in people’s consumption patterns towards more modern ones. With the development of technology that facilitates human life, such as the communication process without thinking about distance, space and time, it is easier to obtain information quickly [2]. One form of technology that is in demand is online shopping. Online shopping makes it easier for someone to shop without spending time and energy; this increases people’s purchasing power [3].

A total of 138.1 million Indonesians aged 18–64 years have used the internet to make purchases through e-commerce and 74.4% have made food purchases through online food delivery applications in East Java, to date. In a study conducted by Nielsen, the food and beverage category was ranked second (42%) as the category of goods most frequently purchased online by Indonesians [4]. The applications that are most frequently used by Indonesian people to purchase food and beverages via online in 2021 are Grab Food (71%) and Go Food (78%) [5].

The online food and beverage shopping application offers the convenience of selecting, ordering, paying and delivery in one application. Most of the foods sold in online applications are foods that are high in carbohydrates, high in fat, and low in fruit and vegetables, so it can be concluded that the food sold is energy dense, nutrition poor [6]. Based on a food trend report in 2021 from Grab Indonesia, the types of food that are most often sought after in Indonesia are fast food and martabak, while the food most ordered by Indonesian people is spicy fried noodles. The KNEKS survey also showed that 27% of Indonesians consume fast food every day and 22% consume it three to four times per week [7]. Consumption of foods that are high in carbohydrates and fats continuously without control can lead to weight gain [8]. If it is not balanced with physical activity, it can be obesity which will trigger the emergence of non-communicable diseases such as hypertension, diabetes mellitus, obesity, and others.

Based on the results of basic health research by the Ministry of the Republic of Indonesia, there was an increase in the prevalence of obesity by 7% from 2013 to 2018 to 21.8% [5]. In addition, the prevalence of diabetes and hypertension also increased to 10.9% and 34.1% in 2018 [7]. The increase in the prevalence of obesity and other diseases can be caused by the behavior and habits of consuming high-calorie foods and beverages [9], besides the low level of physical activity carried out. A person’s eating habits affect their nutritional status. High-calorie food and beverage consumption habits can be influenced by the ease of ordering food through online applications. In addition to the delicious taste, buyers feel a pleasant experience because of how easy it is [10]. Another problem was that it was found that there was no nutritional information relating to the food and beverages sold through the online food and beverage ordering applications, from either Grab Food or Go Food. This not only affects the health of individuals and populations; the causes of obesity problems also have a huge cost not only to the world’s health system but also to a substantial ecological cost to the environment [11].

Preliminary studies that have been conducted on students in the city of Surabaya showed that one of the obstacles in controlling salt and sugar consumption through online applications was the absence of information on salt and sugar content on the online ordering applications. Direct observations on the food delivery applications were also in line with the results of a preliminary study, namely that the information on food and beverages sold on online food delivery applications only gave brief food descriptions.

The health belief model is a model that focuses on efforts to improve public health by understanding why people fail to adopt preventative health measures. This model can be used to explain and predict various positive health-related behaviors [12]. In this model, there are four kinds of beliefs that describe their perception of their health. The first is perceived vulnerability. This perception holds that people will be more motivated to act on healthy behaviors if they believe they are susceptible to negative health outcomes. The second is perceived severity. It argues that the stronger people’s perceptions of the severity of negative health outcomes, the more they will be motivated to act to avoid those outcomes. Third, the barriers and perceived benefits argument suggests that when strong barriers are found in the way of adopting preventative behavior, they are unlikely to do so. Lastly, self-efficacy proposes that overall motivation to pursue health may influence their decision to display positive behaviors.

Based on these problems, this study aims to determine the relationship between the habit of buying food and beverages online from the perspective of health beliefs or the health belief model with the nutritional status of the population in East Java. The role of empowerment and the education system play an important role in encouraging healthy lifestyles, one of which is diet for young people who are prone to overconsumption and obesity [13]. We hope that this research can be developed in the next year for youth health promotion activities in Surabaya and Pasuruan.

## 2. Materials and Methods

This research is a quantitative research study. The quantitative method uses a cross sectional design to determine the association between consumption habits and health beliefs related to the habit of consuming food and drinks purchased online with the nutritional status of the population in East Java. This research was conducted in two big cities in East Java, namely Surabaya and Pasuruan.

Data collection was carried out offline using paper questionnaires from respondents who met the inclusion criteria. The inclusion criteria in this study were adolescents and adults who ordered online food more than once, aged 18–50 years, lived in Surabaya and Pasuruan, Indonesian citizens, and who provided filled-out questionnaires. The exclusion criteria were respondents who never ordered food online or did not provide data. Samples were taken by random sampling until 100 samples were obtained.

This research passed the ethics test through the ethics committee of Universitas Airlangga Surabaya with the ethical number 438/HRECC.FODM/VII/2022. The research questionnaire contained questions about respondents’ characteristics (age, educational background, gender, marriage status, occupation, income level for respondents who are already working, pocket money level for respondents who were still students, domicile status, and calculation of weight and height); the frequency of buying food online; the popular brands of food and beverages they bought most often and questions about Health Belief Model (HBM). Measurements of height and weight were carried out directly by researchers using a digital weight meter and stadiometer. The popular food and beverage brands were analyzed by researchers through the Grab Food or Go Food applications on the most popular food menus in Surabaya and Pasuruan with a minimum rating of 4.5+. Questions regarding HBM were presented with a choice of 1–5 on Linkert scales and were analyzed using statistical methods. The score of each HBM variable was calculated based on the maximum and minimum values and intervals. The research questionnaire also passed the validity and reliability tests conducted in Indonesia with a significance value or *p*-value < 0.05. This questionnaire passed the reliability test with a Cronbach alpha *p*-value > 0.70. The test was carried out using chi square with the help of the IBM SPSS version 20 application.

## 3. Results

This study was conducted on 100 respondents. The majority of the respondents were aged 18–25 years, amounting to 74 people. The majority in this study were female, 85 people, and the males amounted to 15 people. Their educational backgrounds were very diverse with the mean number graduating from high school being 39 people. There were 78 people not married; 65 people were still students. The workers had an income of IDR 2,000,000 or IDR 2,000,000–<4,000,000 per month amounting to 12 people. Meanwhile for students, 17 people had an allowance of IDR 400,000–<600,000 per month. Forty-nine people lived in their parents’ house. Based on body mass index (BMI), the majority of respondents, 64 people, had a normal nutritional status. Characteristics of the respondents are described in the Table 1.

### 3.1. Frequency of Online Orders on Grab Food or Go Food

#### 3.1.1. Frequency of Online Order Based on Mealtime

The results of the data analysis related to the frequency of buying food and beverages online in one month, for 100 respondents in Surabaya and Pasuruan are described in the Table 2.

The results of the study showed that the frequency of ordering food and drinks online was varied. The majority of respondents rarely or never ordered food and drinks online in the morning or breakfast, while for lunch, the majority of respondents thought that they had never ordered food or drinks online. However, the answers to rarely and often consuming food and drinks online were not much different. Respondents rarely bought food and drinks online for lunch and some respondents often consumed food and drinks online for lunch. For snacks during the day, the majority of respondents never ordered food and drinks online through the Grab Food and Go Food application. As with lunch, the frequency of food and drinks ordered for dinner also only had small differences. The numbers of respondents who never bought food and drinks online for dinner were similar to those who bought rarely or often. At mealtimes or snacks after dinner, the majority of respondents never ordered food or drinks online through Grab Food and Go Food applications.

#### 3.1.2. Frequency of Online Order Based on Types of Food and Drink

The frequency of buying food and drinks online measured how much food and drink was purchased using the Grab Food or Go Food applications in the past month. The types of food and drinks selected by the researchers were the 12 popular foods and drinks consumed by users of the application. Table 3 are the results of the distribution of answers from 100 respondents in Surabaya and Pasuruan.

From the table, the information was obtained that the frequency of consuming food and drinks purchased on Grab Food or Go Food was very diverse. For the types of drinks, the majority of respondents consumed boba drinks 1–5 times a month. As for soft drinks, the majority of respondents had never bought them through online applications. Similar to boba drinks, most respondents bought drinks containing coffee 1–5 times a month. However, the others never ordered them online. As for the type of food, the majority of respondents often consumed various types of processed chicken, noodles, and meatballs 1–5 times a month. As for various fried foods, vegetables and fruit, the majority of respondents did not buy them online.

### 3.2. Health Belief Model 

#### 3.2.1. Perceived Susceptibility

Perceived susceptibility is an individual’s belief about his or her susceptibility to disease risk in encouraging people to adopt healthier behaviors. The question asked was about the respondent’s perception of their perceived vulnerability when consuming food and beverages purchased from Grab Food or Go Food. The following are the results of the distribution of answers from 100 respondents.

Based on Table 4, the majority of respondents agreed that consuming foods with high calories could cause health problems related to degenerative diseases such as diabetes mellitus, hypertension, and obesity. Most of the respondents also agreed that they consumed drinks high in sugar, salt, other foods with high calories, had low physical activity, and often consumed junk food and foods that are low in vitamins and minerals would make a person susceptible to degenerative diseases and make the body unfit. However, there were balanced results amounting to 50 respondents agreeing and 50 respondents doubting that the need for nutritional information on the food and beverages consumed could make a person susceptible to obesity. In contrast to the results of the answers relating to the availability of nutritional information, the majority of respondents agreed that if there were nutritional information, they would feel protected. Another thing, that the majority of respondents agreed, was that the ease of ordering food and drinks online made them become consumptive individuals.

#### 3.2.2. Perceived Severity

Perceived severity is the feelings about the seriousness of degenerative disease including the evaluation of clinical and medical consequences (such as death, disability, and illness) and the possible social consequences (such as effects on work, family life, and social relationships). Many experts combine the two components above as a perceived threat.

Based on Table 5, the majority of respondents agreed that they were too lazy to exercise and do physical activity is a risk for obesity. The respondents also agreed that if they were overweight or obese, it would be difficult to lose the weight. Most of the respondents also agreed that if they had some signs or symptoms of the degenerative diseases that can cause difficulties in carrying out daily activities, they would shorten their life expectancy, and give risk to future generations. The respondents also agreed that if they had degenerative diseases such as diabetes mellitus, hypertension, and obesity, they would be troublesome for their families and have difficulty in paying for treatment. However, the respondents also agreed that if they had pre-diabetes it would cause them to feel vulnerable to developing diabetes in the next 2–5 years. Furthermore, the majority of respondents agreed that if they suffered from early hypertension it could cause them to feel vulnerable to heart disease and stroke.

#### 3.2.3. Perceived Benefit

Perceived benefit is the acceptance of a person’s susceptibility to a condition that is believed to cause serious risk (perceived threat) which encourages them to produce a force that supports changes in nutritional behavior. This depends on a person’s belief in the effectiveness of the various available means in reducing the threat of degenerative disease, or the perceived benefits of making these health changes. When a person shows a belief in susceptibility and seriousness, he or she is often not expected to accept any recommended health measures unless they are felt to be appropriate and effective.

Based on Table 6, the majority of respondents agreed that their diet helped keep them to stay healthy and strong in daily activities. The respondents also agreed that cooking for themselves at home could save expense. Most of the respondents were also neutral about eating healthy food according to their plate (half the plate for vegetable and fruit, on quarter for the side dish, and one quarter for carbohydrate) in order to maintain ideal body weight. However, the respondents did not agree that they felt cool (did not miss it) when buying up-to-date food and drink on Grab Food or Go Food. However, the respondents were also neutral about not eating food or sweet drinks, salty food, and coconut milk or fatty foods purchased from Grab Food or Go Food to prevent the risk of developing diabetes, hypertension, and stroke. Another thing that the majority of respondents agreed upon was that when they ate healthy food or drinks and fruit or vegetables which were purchased from Grab Food or Go Food, it made their bodies feel healthier and increased their immune systems. However, the respondents were also neutral when they ate animal or vegetable side dishes purchased from Grab Food or Go Food about whether they could increase their immune system.

#### 3.2.4. Perceived Barrier

A perceived barrier is the potential negative aspect of a prevention and treatment effort in dealing with degenerative disease (such as uncertainty and side effects), or perceived barriers (such as worrying about being unsuitable, unhappy, and nervous), which may serve as barriers to recommending a behavior.

Based on Table 7, the majority of respondents agreed that they did not have enough time to cook so bought food from Grab Food or Go Food. The respondents also agreed that healthy food and drink options (salads, less sugar, low fat drinks) sold by Grab Food or Go Food were more expensive. Most of the respondents also agreed that they were too lazy to leave the house to buy food or drinks so bought from Grab Food or Go Food. The respondents also agreed that there was no information about the protein, fat, carbohydrate, sugar, and salt content of the food sold on the Grab Food or Go Food applications. However, the respondents also agreed that there were no low-sugar food or drink choices on the Grab Food or Go Food applications. Furthermore, the majority of respondents were neutral over the high-protein foods or drinks sold by Grab Food or Go Food being less varied.

#### 3.2.5. Perceived Cues to Action

Perceived cues to action are signs that exist in an individual’s life that can support and encourage them to perform health behaviors. Based on the results of a survey of 100 respondents, information was obtained which is described in the table below.

Based on Table 8, the majority of respondents agreed they were influenced by other friends to buy food and drinks sold by Grab Food or Go Food. The majority of respondents also agreed that they often received a number of promos and discounts when they placed an order. This gave them the desire to order food or drinks sold online. However, the majority of respondents felt neutral about the way they found out about the nutritional content and nutritional composition before buying something. The majority of respondents felt neutral about the aspect of learning at cooking demonstrations in order to increase their own knowledge and cooking skills at home. However, the majority of respondents agreed that they spent time cooking with their families. In addition, the majority of respondents agreed that when they ate, they provided vegetables and made balanced food components. Unfortunately, the majority of respondents had not used or were neutral to familiarizing themselves with nutrition applications; the majority of respondents had not joined a sports community and did not have an appropriate exercise schedule.

#### 3.2.6. Self-Efficacy

Self-efficacy is a belief in a person’s ability to take an action related to balanced nutrition consumption and physical activity in order to achieve a goal in preventing and dealing with degenerative diseases.

Based on Table 9, the majority of respondents agreed that knowing the nutritional content of the food and beverages they consumed would enable them to maintain their health. The majority of respondents also agreed they could find out the benefits of what they consumed. However, almost half of the respondents felt neutral about making their own food and drink every day with the hope of reducing online food and beverage purchases. The majority of respondents also agreed that they could control the consumption of food and beverages purchased online even though there were many promotional offers. The majority of respondents also agreed they could find out the nutritional content of the items they consumed. The majority of respondents were also neutral about prioritizing health over desire when consuming food and beverages purchased online. Similarly, the attitude of respondents who believed that they were able to take 15–20 min to exercise. The majority of respondents also agreed that they could find out the composition of the nutritional content of what they consumed. Unfortunately, the neutral belief in the respondent’s ability to maintain an ideal body weight differed slightly from the answer of agreement. Meanwhile, the majority of respondents agreed that they were selective in consuming food and beverages before buying.

### 3.3. Association between Characteristics and Health Belief

There was an association between age, last education level, marriage status, profession, income per month (for workers), allowance per month (for students), and health beliefs. However, there was no association between gender, domicile status, weight, or height, and health beliefs.

Based on Table 10, there was a significant association between age and health beliefs. The significant association is shown by the *p* value < 0.05. The significance association also occurred with other characteristics such as last education level, marriage status, profession, income per month (for workers), allowance per month (for students), and health beliefs. However, there was no significant association between gender and health beliefs. The insignificant association is shown by the *p* value > 0.05. The insignificance association also occurred with other characteristics, such as domicile status and BMI, and health beliefs.

### 3.4. Association between Health Belief and Frequency of Online Order

There was no association between health beliefs and frequency of online order based on mealtime, such as breakfast, lunch, snacks in the afternoon, dinner, and snacks in the evening.

Based on Table 11, there was no significant association between health beliefs and frequency of online order and breakfast. The insignificant association is shown by the *p* value > 0.05. The insignificance association also occurred for other mealtimes such as lunch, snacks in the afternoon, dinner, and snacks in the evening. However, health beliefs are 4535 times more likely to affect online orders for food and drinks at breakfast-time. Furthermore, health beliefs are 0.776 times more likely to affect online orders for food and drinks at lunchtime. Health beliefs are also 3446 more likely to affect the frequency of online orders for snacks in the afternoon. Health beliefs are 5330 times more likely to affect online orders for food and drinks for dinner. Finally, health beliefs are also 5157 times more likely to affect the frequency of online orders for snacks in the evening.

### 3.5. Association between Health Belief and Nutritional Status

Most of the respondents in this study had average scores in perceived benefit, cues to action, and self-efficacy, but scored high in perceived susceptibility, perceived severity, and perceived barriers. There was no association between perceived susceptibility, perceived severity, perceived benefit, perceived barrier, cues to action, or self-efficacy, and nutritional status.

Based on Table 12, there is no significant association between perceived susceptibility and nutritional status. The insignificant association is shown by the *p* value > 0.05. The insignificance association also occurred with other independent variables, such as perceived severity, perceived benefit, perceived barrier, cues to action, self-efficacy, and nutritional status. However, perceived susceptibility showed that those who had high perceived susceptibility were 4256 times more likely to see an effect on nutritional status. Another variable, perceived severity, showed that those who had high perceived severity were 9803 times more likely to see an effect on nutritional status. Perceived benefit respondents were also 2358 more likely to see an effect on nutritional status. Perceived barrier showed that those who had a high perceived barrier were 2993 times more likely to affect nutritional status. Cues to action respondents are also 5814 more likely to see an effect on nutritional status. The last independent variable, self-efficacy, showed that those respondents were also 3644 times more likely to see an effect on nutritional status.

## 4. Discussion

Health is a condition that is not only free from disease, but also covers all aspects of human life which include the physical, emotional, social and spiritual. According to the WHO (1974), health can be defined as a state of complete physical, mental and social well-being and not merely the absence of disease or infirmity. In the broadest sense, health is a dynamic state in which individuals adapt to changes in the internal (psychological, intellectual, spiritual and disease) and external (physical, social and economic) environments in order to maintain their health [14].

Factors that affect human health have been described in various theories and research. Among these, is the health belief model. The health belief model is a model that focuses on efforts to improve public health by understanding why people fail to adopt preventative health measures. This model can be used to explain and predict various positive health-related behaviors [12]. HBM theory has been widely used in various studies to assess individual beliefs about a behavior related to health. The HBM contains several primary concepts that predict why people will take action to prevent, to screen for, or to control illness conditions; these include susceptibility, seriousness, benefits and barriers to a behavior, cues to action, and most recently, self-efficacy. If individuals regard themselves as susceptible to a condition, believe that a condition could have potentially serious consequences, believe that a course of action available to them would be beneficial in reducing either their susceptibility to or the severity of the condition, and believe the anticipated benefits of taking action outweigh the barriers to (or costs of) action, they are likely to take action that they believe will reduce their risks. In the case of medically established illness (rather than mere risk reduction), the dimension has been reformulated to include acceptance of the diagnosis, personal estimates of susceptibility to consequences of the illness, and susceptibility to illness in general [15].

The studies conducted by Hochbaum, in 1952, related to a prevention program against tuberculosis, were fundamental for the development of the HBM [16]. These studies observed more than 1200 adults in three American cities and their willingness to undergo X-ray examinations. They found that their willingness to undergo examinations was the product of an individual belief of susceptibility to the disease and the personal benefits of early detection [16]. Another study from India explained that the model of health belief could be used to change people’s perspectives and materials in diabetes mellitus education. The educational training intervention in the HBM model showed a significant increase in self-care behaviors in the intervention group as compared to the control group [17]. The HBM had a significant impact on increasing the scores of awareness, perceived susceptibility, severity, benefits, and self-efficacy, cues to action and preventative behaviors and on decreasing the score of perceived barriers among students. The health education interventions in the HBM were practical approaches in educating and promoting proper health behaviors concerning type 2 diabetes among students in Kash (southwestern Iran) [18]. The HBM was proposed, at first, to give an explanation and prediction of preventative behaviors and to find out the reasons for people not going to medical examinations for early detection of diseases or simply to know their health status, among other forms of preventative behaviors [19].

As previous studies explained, there are many roles for the HBM in various aspects. Some of them are used to analyze individual health perspectives through six HBM variables, or use them as models for an educational approach to change individual habits and behavior. This explains that health belief is often an independent variable in various studies. In addition to analyzing the relationship between health beliefs related to food and beverage consumption habits when purchasing online through Grab Food and Go Food, this study tried to analyze the relationship between demographic factors and health beliefs. As a result, several variables were interrelated. The variables are age, education level, marriage status, profession, income, and monthly allowance

Age can affect a person’s health belief. The older a person gets, the greater their confidence in their health will be. Several studies explain the same thing. Research conducted using online surveys on apps (Twitter, Facebook and LinkedIn) on residents of Ontario, Canada explained that middle-aged and older adults reported greater concern about the personal risks of hospitalization and death, the economic and social impact of COVID-19 than young adults [20]. Structural equation modeling suggests the perceived benefits of health behaviors are the main drivers of behavioral uptake, with socioeconomic factors and perceived severity and vulnerability being indirectly related to uptake through their influence on perceived benefits [20]. Another study explained that age was related to biological and ecological aspects, which included the impact of human physiology on their needs, purchasing behavior and consumption [21]. Because of that, features can be extracted that have a particular impact on the nutritional needs, such as age, height, weight and physical fitness [21].

Education level. Education is a means that allows a person to acquire a variety of useful knowledge about aspects of life, one of which is health. In this study, there was a significant relationship between education level and health beliefs related to the habit of consuming food and beverages purchased online using the Grab Food and Go Food applications. The higher the level of education, the higher the health belief. This is in line with research conducted by Tiziano that the level of education can affect a person’s knowledge and prevent someone from believing there is a virus related to health problems, such as COVID-19 [22].

Marriage status. This study explains that marriage can affect a person’s health, especially in terms of the habit of consuming food and drinks ordered through the Grab Food and Go Food applications, as well as a healthy lifestyle. Most of the respondents often cook themselves at home with the aim of reducing costs due to buying food and drinks through online applications. This is in line with Backer’s statement, In his influential work through specialization and economies of scale, marriage causes an increase in the resources of a married couple, thereby increasing their economic wealth [23]. There have been many studies related to marital status and health. The first group of studies to link better health and marriage used economics to explain this relationship. Roher et al. (2008) found in their clinical work on women that being single, over the age of 65, having more physical symptoms than most patients, and feeling depressed were each independently associated with a lower self-assessment of health status [24].

Profession. A person’s profession or type of work can affect their beliefs and their healthy behavior, especially in terms of the habit of consuming food and beverages purchased through online Grab Food and Go Food applications. Researchers assume that working hours and workload can affect the habit of buying food and drinks online. Someone will choose a practical way by ordering online [25]. This assumption is driven by the answer to the questionnaire that the majority agreed that they ordered food and drinks online because they did not have enough time to cook for themselves at home. In this type of profession, people often have close colleagues and friends. Colleagues and close friends influence the cue to action ordering food and drinks online.

Income and allowance. Income and allowances can affect beliefs and perspectives on health, especially related to the habit of consuming food and beverages purchased online through the grab food and go food applications. Researchers assume that the availability of money can be an enabling factor that makes someone consider buying food and drinks online. The availability of more money to meet the need for practical food and drinks makes a person more vulnerable to consuming food and drinks purchased online. This assumption is in line with the explanation of C. Bywalec and L. Rudnicki that one of the factors that influences food-buying behavior is the economic factors, among which are: the resources, and the availability of food, household income and the percentage of income spent on food, the level and the relationship of price of consumer goods, the supply of consumer goods; there is also the impact of environment, size and following the standard model [21].

In addition to testing the characteristics of the respondents’ demographic factors on health beliefs, the researchers examined the relationship between health belief and order frequency. However, there is no significant relationship between the two variables. Several previous studies have explained that the frequency of eating can affect health problems that have already occurred. Research conducted in Indonesia on students at SMA Kartini Batam, in 2018, showed 22 respondents with a percentage (42.3%) whose frequency of eating junk food was rare and 28 respondents with a percentage (53.8%) had normal nutritional status. Based on the correlation test, it was found that there was a significant relationship between the frequency of eating junk food and nutritional status in students with *p* value 0.000 [26].

The more often a person eat junk food, the more it will affect their nutritional status and they may experience overweight or obesity. Previous research explained that when we refer to food, overconsumption means overeating, and it is the situation in which an individual consumes food above the body’s energy requirements in relation to energy expenditure, leading to excess fat in the body. When practiced constantly for long periods of time, and coupled with lack of physical activity, overeating is the principal cause of overweight and obesity in both adults and children [27].

Finally, this research found no significant relationship between health belief and nutrition status. The question posed about perceived susceptibility concerned the relationship between behavior and habits of consuming food and beverages purchased online in Grab Food and Go Food, as well as behaviors relating to a healthy lifestyle. In this study, most of the respondents were in the score of as much as 65% of perceived vulnerability. As many as 65% of respondents with high susceptibility beliefs had normal nutritional status. This was to be expected because belief makes a person more certain of a healthy lifestyle. One of the behaviors is food consumption with exercise.

Perceived severity or perceived seriousness concerns perceptions or beliefs that are felt about the serious impact of the problem due to an unhealthy lifestyle and consumption patterns of foods and beverages that contain a lot of calories and a person with insufficient physical activity [16]. This includes clinical and medical consequences (e.g., poor health, illness, and death) and possible consequences (such as effects on work, family life, and social relationships). Many experts combine the above two components as a perceived threat. In this study, most of the respondents had a high or good perception of seriousness (69%). The respondents with a high perception of seriousness had normal nutritional status (45%), although 16% and 2% of them were overweight and obese, respectively. The seriousness of the impact can be influenced by the level of understanding and knowledge about the dangers of degenerative diseases that can be caused.

Perceived benefit is the acceptance of a person’s vulnerability to a condition that is believed to cause serious risk (perceived threat) which encourages them to produce a force that supports changes in nutritional behavior [28]. The effectiveness of various existing efforts in reducing the threat of degenerative diseases, or the perceived benefits of carrying out these health efforts depends on the person. Among these health efforts are doing physical activity for 30 min 3–4 times a week, consuming a balanced diet according to the “fill my plate” program and reducing sugar, salt and fat consumption. When a person believes in the expected vulnerability and seriousness, it is often not possible to accept the prescribed health action unless it feels appropriate and relevant. As a result, the majority of respondents have a belief or perception of the benefits in the moderate category (53%). Respondents who are in accordance with this moderate belief have normal nutritional status. This could be due to the lack of knowledge related to the perceived benefits.

Perceived barriers are potential negative aspects or perceived barriers of the effort leading to healthy behavior and a healthy lifestyle [29]. In this study, the researchers examined the perceived barriers when choosing to buy food and beverages online through the Grab Food and Go Food apps. However, the applications are not yet equipped with nutritional information to give users the opportunity to choose healthier foods for them. As a result, the majority of respondents have high confidence in obstacles (74%). Respondents felt they had many obstacles when they wanted to carry out healthy behaviors including the behavior of ordering food and drinks online.

Cues to action are anything that can encourage and support behavior. In this study, emphasis was placed on matters relating to activities, actions and facilities that could encourage a person to carry out healthy behavior and a healthy lifestyle, including how they ensured their nutritional status was good and normal. From the results, it was found that the majority of respondents had signs for moderate cues to action (47%) with the majority of them having normal nutritional status. This indicates that more efforts are needed to present those things related to behavioral drivers, with the hope of creating optimal healthy behavior.

Self-efficacy is the belief in one’s ability to take an action to achieve a goal in carrying out a healthy lifestyle and behavior. In this study, most of the respondents had a moderate score. This means that respondents had moderate confidence in carrying out healthy behavior, including when ordering and choosing types of food and drinks from Grab Food and Go Food. Based on the results of the analysis using SPSS, information was obtained that there was no relationship between each HBM variable and the high nutritional status of the community in Surabaya and Pasuruan. This could be caused by external factors including the local social environment.

External factors that can affect a person’s nutritional status include the variety of food consumed and the environment in which they live. The total energy consumption, fat consumption, snack frequency, fast food consumption frequency, snacking habits while watching TV, between physical activity, watching TV, sleeping, parental education level, mother’s work status, knowledge of maternal nutrition, all have a significant relationship to the nutritional status of the child [30]. Community education and empowerment is needed to increase their awareness around understanding their needs and developing ways to access healthy and fresh food. It aims to increase consumer awareness of a healthy and sustainable diet, along with a uniform distribution system of nutritious food around the world and reduce food waste due to overconsumption [27]. This effort is part of a global effort to manage the world’s economy and food resources [27].

## 5. Conclusions

Most of the respondents in this study scored average in perceived benefit, cues to action, and self-efficacy. However, they scored high in perceived susceptibility, perceived severity, and perceived barriers. There is a significant association between the characteristics of the respondents (age, marriage status, profession, education level, income, and allowance) and their health beliefs. There is no association between health beliefs and frequency of online order. Finally, no significant association was found between perceived susceptibility, perceived severity, perceived benefit, perceived barrier, cues to action, or self-efficacy and nutritional status. Thus, it is still important for the government to increase socialization and education in the importance of balanced nutrition and nutritional status so that the people can protect themselves and prevent the onset of degenerative diseases.

## Figures and Tables

**Table 1 nutrients-14-04482-t001:** Characteristics of the respondents.

	Category	%
Age	18–25 years old	74
	26–30 years old	10
	31–25 years old	5
	36–40 years old	6
	41–45 years old	3
	46–50 years old	1
	51–55 years old	1
	56–60 years old	0
Gender	Male	15
	Female	85
Last Education Level	Uneducated	1
	Primary school graduate	0
	Secondary school graduate	26
	High school graduate	39
	Diploma graduate	3
	University graduate	31
Marriage Status	Not married	78
	Married	21
	Divorced	1
Profession	Student	65
	Government employees	19
	Private employees	9
	Freelance	1
	Other	6
Income per Month (for Workers)	IDR < 2,000,000	12
	IDR 2,000,000–<4,000,000	12
	IDR 4,000,000–<6,000,000	5
	IDR 6,000,000–<8,000,000	4
	IDR > 8,000,000	3
Allowance per Month (for Students)	IDR < 200,000	7
	IDR 200,000–<400,000	15
	IDR 400,000–<600,000	17
	IDR 600,000–<800,000	11
	IDR > 800,000	14
Domicile Status	Own house	18
	Parent’s house	49
	Relative’s house	1
	Cost/contract	29
	Other	3
Body Mass Index (BMI)	Underweight	10
	Normal	64
	Overweight	16
	Obese	10

**Table 2 nutrients-14-04482-t002:** Frequency of online order based on mealtime.

	Category	%
Breakfast	Never	77
	Rarely/sometimes	15
	Often	8
Lunch	Never	38
	Rarely/sometimes	34
	Often	28
Afternoon Snack	Never	51
	Rarely/sometimes	37
	Often	12
Dinner	Never	39
	Rarely/sometimes	37
	Often	24
Evening Snack	Never	63
	Rarely/sometimes	31
	Often	6

**Table 3 nutrients-14-04482-t003:** Frequency of online orders based on types of food and drink.

	Category	%
Boba Drink/Milk Tea/Thai Tea	0	40
	1–5	57
	6–10	0
	11–15	1
	≥16	0
Soft Drink	0	80
	1–5	19
	6–10	1
	11–15	0
	≥16	0
Coffee	0	50
	1–5	44
	6–10	5
	11–15	1
	≥16	0
Fast Food	0	41
	1–5	54
	6–10	4
	11–15	1
	≥16	0
Chicken	0	18
	1–5	71
	6–10	8
	11–15	3
	≥16	0
Rice	0	33
	1–5	54
	6–10	11
	11–15	2
	≥16	0
Sweet Martabak/Toast	0	50
	1–5	48
	6–10	1
	11–15	0
	≥16	1
Salty and Spicy Snacks	0	44
	1–5	49
	6–10	5
	11–15	2
	≥16	0
Noodles and Meatballs	0	38
	1–5	55
	6–10	7
	11–15	0
	≥16	0
Fried Food	0	49
	1–5	42
	6–10	5
	11–15	4
	≥16	0
Vegetables	0	63
	1–5	28
	6–10	7
	11–15	1
	≥16	1
Fruit	0	59
	1–5	31
	6–10	8
	11–15	1
	≥16	1

**Table 4 nutrients-14-04482-t004:** Perceived susceptibility results.

Perceived Susceptibility Questions	Percentage of Respondents with Agreement Level				
	Strongly Disagree (%)	Do Not Agree (%)	Neutral (%)	Agree (%)	Strongly Agree (%)
If I eat high-calorie food, I feel vulnerable to health problems (hypertension, diabetes mellitus, obesity, etc.).	8	9	22	45	16
If I consume sweet drinks that are purchased from Grab Food/Go Food and contain more than 4 tablespoons of sugar, I feel susceptible to getting hit by hyperglycemia.	7	10	21	52	10
If there is nutritional information (content of fat, sugar, protein, carbohydrates, etc.) about the drinks and food consumed, I feel safe.	1	4	24	44	27
If I go for the convenience of ordering food from Grab Food/Go Food, I feel prone to become a consumptive individual.	0	9	29	43	19
If I do not know the nutritional content of the food or drink which I consume, I feel prone to becoming overweight or obese.	2	17	31	44	6
If I eat salty food purchased from Grab Food/Go Food which contains more than 1 teaspoon of salt, I feel susceptible to hypertension	5	11	34	43	7
If I do not do physical activity, I’m at risk of becoming overweight or obese.	2	14	18	47	19
If I eat high-calorie foods without adequate physical activity, I feel that my weight will increase.	1	6	14	48	31
If I often eat fast food or junk food, it will affect my health.	0	3	19	51	27
If I eat food high in vitamins and minerals such as vegetables and fruit every day, I feel that my body is fit and healthy.	0	0	14	48	38

**Table 5 nutrients-14-04482-t005:** Perceived severity results.

Perceived Severity Questions	Percentage of Respondents with Agreement Level				
	Strongly Disagree (%)	Do Not Agree (%)	Neutral (%)	Agree (%)	Strongly Agree (%)
If I am too lazy to exercise and do physical activity, I am at risk of obesity.	1	9	16	53	21
If I am overweight or obese, I find it difficult to lose weight.	1	18	27	34	20
If I have some signs or symptoms of a disease that appears degenerative (diabetes, hypertension, stroke, etc.), I find it difficult to carry out daily activities.	3	8	19	57	13
If I have some signs or symptoms of a disease that appears degenerative (diabetes, hypertension, stroke, etc.), I feel like I cannot live much longer.	8	30	23	31	8
If I have a degenerative disease (diabetes, hypertension, stroke, etc.), I feel it will be troublesome for my family.	4	8	8	59	21
If I have a degenerative disease (diabetes, hypertension, stroke, etc.), I will find it difficult to pay for treatment.	4	4	23	53	16
If I have pre-diabetes, I feel vulnerable to developing diabetes in the next 2–5 years.	3	9	21	54	13
If I suffer from early hypertension, I feel vulnerable to suffering from heart disease and stroke.	2	10	22	56	10
If I have some signs or symptoms of degenerative disease, I will pass the risk on to generations or my descendants.	4	8	24	46	18

**Table 6 nutrients-14-04482-t006:** Perceived benefit results.

Perceived Benefit Questions	Percentage of Respondents with Agreement Level				
	Strongly Disagree (%)	Do Not Agree (%)	Neutral (%)	Agree (%)	Strongly Agree (%)
I keep my diet to stay healthy and strong in daily activities.	0	3	23	51	23
I cook for myself at home so I can save expenses.	1	4	26	53	16
I eat healthy food according to my plate (half the plate for vegetables and fruit, one quarter for the side dish and one quarter for other carbohydrates) in order to maintain ideal body weight	0	10	41	39	10
I feel cool (do not miss it) when buying up-to-date food and drink on Grab Food/Go Food.	7	44	34	12	3
I do not eat food or sweet drinks that are purchased from Grab Food/Go Food to prevent the risk of developing diabetes.	2	27	39	31	1
I do not eat salty food that is purchased from Grab Food/Go Food to prevent the risk of developing hypertension	1	27	45	25	2
I do not eat coconut milk or fatty foods that are purchased from Grab Food/Go Food to prevent the risk of stroke.	2	24	34	33	7
When I eat food or healthy drinks that are purchased from Grab Food/Go Food, then my body feels healthier.	0	11	33	44	12
When I eat fruit or vegetables that are purchased from Grab Food/Go Food, then my immune system improves.	0	10	39	47	4
When I eat animal or vegetable side dishes that are purchased from Grab Food/Go Food, then my immune system improves.	0	7	46	41	6

**Table 7 nutrients-14-04482-t007:** Perceived barrier results.

Perceived Barrier Questions	Percentage of Respondents with Agreement Level				
	Strongly Disagree (%)	Do Not Agree (%)	Neutral (%)	Agree (%)	Strongly Agree (%)
I do not have enough time to cook, so I buy food from Grab Food/Go Food.	2	20	29	40	9
Healthy food and drink options (salads, less sugar, low fat drinks) sold by Grab Food/Go Food are more expensive.	0	10	31	44	15
I am too lazy to leave the house to buy food or drinks so I buy from Grab Food/Go Food.	0	10	17	59	14
There is no information about the protein content of the food sold on the Grab Food/Go Food applications.	1	9	27	44	19
There is no information about the fat content in food and beverages sold on the Grab Food/Go Food applications.	1	12	24	41	22
There is no information about the carbohydrate content of foods and beverages sold on the Grab Food/Go Food applications.	1	9	28	42	20
There is no information about sugar content in the food and beverages sold on the Grab Food/Go Food applications.	1	11	30	38	20
There is no information about the salt content in the food and beverages sold on the Grab Food/Go Food applications.	1	12	29	37	21
There is no choice of low-sugar foods or drink on the Grab Food/Go Food applications.	1	21	30	36	12
High-protein foods or drinks sold by Grab Food/Go Food are less varied.	1	21	38	34	6

**Table 8 nutrients-14-04482-t008:** Perceived cues to action results.

Perceived Cues to Action Questions	Percentage of Respondents with Agreement Level				
	Strongly Disagree (%)	Do Not Agree (%)	Neutral (%)	Agree (%)	Strongly Agree (%)
My friends or colleagues often invite me to buy food or drinks from Grab Food or Go Food when they are together (meetings, study assignments, relaxing, etc.)	0	16	15	55	14
I am often tempted by promos or discounts (price cuts, free shipping, cashback, etc.) on Grab Food or Go Food	0	8	18	53	21
I look for information about the ingredients (wheat flour, eggs, milk powder, soy sauce, sugar, salt, etc.) in a food or drink before buying it.	2	33	42	19	4
I look for information about the nutritional content (energy, carbohydrates, protein, and fat) in a food or drink before buying it.	2	28	48	18	4
I have attend cooking demonstrations to increase my knowledge and skills.	4	27	43	23	3
I take time to cook with my family.	1	11	28	53	7
I provide staple foods, side dishes, vegetables, fruit, and drinks at every meal.	0	17	36	42	5
I use certain applications to find out the nutritional intake of the food or drink that I consume in a day.	4	34	41	19	2
I am a member of a sports community (gymnastics, cycling, basketball, football, etc.)	6	42	24	19	9
I have a regular exercise schedule	3	34	28	26	9

**Table 9 nutrients-14-04482-t009:** Self-efficacy results.

Perceived Self-Efficacy Questions	Percentage of Respondents with Agreement Level				
	Strongly Disagree (%)	Do Not Agree (%)	Neutral (%)	Agree (%)	Strongly Agree (%)
I know the level of nutritional adequacy that must be met to keep the body healthy and fit.	0	23	28	47	2
I can find out the benefits of food or drink to be consumed.	0	11	22	64	3
Making your own food or drink is an easy thing to do every day	1	15	36	43	5
I was able to avoid high-calorie drinks sold by Grab Food or Go Food even though they were on promotion.	1	22	25	42	10
I can find out the nutritional content of the food or drink that I want to consume.	2	14	38	41	5
I prioritize the health impact over the desire just before consuming food or drink.	1	13	42	40	4
I am able to spend 15–20 min a day doing physical activity.	1	15	33	34	17
I can find out the composition of the ingredients in the food or drink that I want to consume.	0	15	39	44	4
I am able to maintain my ideal weight.	2	25	31	35	7
I am selective in buying the food or drinks that I consume even though I really want to buy them.	2	9	34	46	9

**Table 10 nutrients-14-04482-t010:** Association between characteristics and health belief.

Characteristics	*p* Value	OR	95% CI	
			Lower	Upper
Age	0.006	17,940	0.000	0.030
Gender	0.119	2427	-	-
Last Education Level	0.000	23,703	0.000	0.030
Marriage Status	0.001	13,408	0.000	0.030
Profession	0.000	20,104	0.000	0.030
Income per Month (for workers)	0.001	21,720	0.000	0.030
Allowance per Month (for students)	0.000	25,149	0.000	0.030
Domicile Status	0.263	5248	0.174	0.346
BMI	0.169	5043	0.041	0.159

**Table 11 nutrients-14-04482-t011:** Association between health beliefs and frequency of online order.

Frequency of Online Order	*p* Value	OR	95% CI	
			Lower	Upper
Breakfast	0.104	4.535	0.013	0.107
Lunch	0.679	0.776	0.463	0.657
Afternoon Snack	0.179	3.446	0.064	0.196
Dinner	0.070	5.330	0.000	0.047
Evening Snack	0.076	5.157	0.007	0.093

**Table 12 nutrients-14-04482-t012:** Association between health belief and nutritional status.

HBM Variable	*p* Value	OR	95% CI	
			Lower	Upper
Perceived Susceptibility	0.642	4256	0.525	0.715
Perceived Severity	0.133	9803	0.130	0.290
Perceived Benefit	0.884	2358	0.699	0.861
Perceived Barrier	0.810	2993	0.756	0.904
Cues to Action	0.444	5814	0.362	0.558
Self-Efficacy	0.724	3644	0.514	0.706

## Data Availability

The datasets used and/or analyzed during the current study are available from the corresponding author upon reasonable request.

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
