# Peer review of "Food and Beverage Consumption Habits through the Perception of Health Belief Model (Grab Food or Go Food) in Surabaya and Pasuruan"

_nutrients, 2022, doi:10.3390/nu14214482_

Round 1

Reviewer 1 Report

This study analyzes the consumption of food and beverages purchased online and the nutritional status of Indonesians. Power and sample size estimation was not assessed.

1-The topic is interesting, but the poor English with errors in spelling, grammar, and punctuation prevented a full understanding of the topic covered. See, for example, on page 3 the sentence “Perceived benefit is [12] Acceptance of a person's vulnerability to a condition that is believed to cause seriousness (perceived threat) is to encourage him to produce a force that supports  changes  in  nutritional  behavior.”

Numerous repetitions are also reported.

2-The tables are all to be revised as it makes little sense to use both absolute numbers and percentages when the number of participants is equal to 100.

3- It should be clearly stated in the Materials and Methods that stature and weight measurements were self-reported by the participants.

4- Table 1 should be divided into two different tables as anthropometric traits and indices are to be reported separately and by sex.

5- In Table 3 and the following, categories 16 to 30 should be unified by simply indicating ≥16.

6- I suggest not commenting on the tables by reporting the same values already given in the table.

7- Sex should be taken into account in the association between Health Belief Model and Nutritional Status.

Author Response

thanks for the review.
1. Thank you for the correction for the use of English in this article. but we need more time maybe about 1 week to make sure the English we use can be better. Therefore, we have not made a maximum revision for this first point.

2. It is suitable
3. We have revised according to your input, please check again on our latest article
4. we don't understand what you mean in the 4th point comment
5. It is suitable
6. What do you think we should show to explain the table?

Thank you so much for your help

Reviewer 2 Report

The manuscript examines the actual problem.

However, the authors misuse the tools of mathematical statistics. Confidence Limits for the Mean are not specified in the methodology. Likewise, the reliability indicators of the difference are not presented and analyzed in the Results section.

The manuscript requires substantial revisions.

Sincerely.

Reviewer 3 Report

The work is relevant for the field and well designed. Congratulations for your work! But requires some corrections and additions. 

1. There are errors in written expression, and for some I gave examples of correction in the comments. A professional proofreading of the entire text is necessary.

2. The abstract must be reformulated. It is presented not in accordance with MDPI requirements, and I wrote the concrete observations in the comment (without numbering, without section titles in the abstract).

3. There are a lot of data and they are difficult to follow in the current presentation. The graphic interpretation of these data and/or the results of their textual interpretations is necessary.

4. References are extremely limited. Only 13 bibliographic references cannot support such a work, if the authors want to produce a work good enough to be published in this journal. The work could increase in value a lot if it cited many more bibliographic sources and presented comparative elements that would support or contradict certain ideas. I recommend expanding the list of bibliographic sources, with the following bibliographic titles, but these are not exhaustive and can be completed, in the interest of supporting the value of the work. My recommendation:

- Serafini, M.; Toti, E. Unsustainability of Obesity: Metabolic Food Waste. Front. Nutr. 2016, 3, 40. Available online: https://www.frontiersin.org/article/10.3389/fnut.2016.00040

- Balan I M, Gherman E D, Brad I, Gherman R, Horablaga A, Trasca TI, 2022, Metabolic Food Waste as Food Insecurity Factor—Causes and Preventions, FOODS MDPI https://www.mdpi.com/1738022

Prescott,M.P.; Burg, X.;Metcalfe, J.J.; Lipka, A.E.; Herritt, C.; Cunningham-Sabo, L. Healthy Planet, Healthy Youth: A Food SystemsEducation and Promotion Intervention to Improve Adolescent Diet Quality and Reduce Food Waste. Nutrients 2019, 11, 1869.

5. The authors have to declare if they are or not in conflict of inters.

Round 2

Reviewer 1 Report

The authors partially revised their manuscript. In particular, in addition to revising the English, I suggest again considering the following concerns:

·       Tables 1 and 2 should be simplified by leaving only the percentage column. Column n is identical (sample =100 participants) and should be eliminated.

·       I suggest not reporting the same data in both tables and text to avoid duplication. In the text you can limit yourself to the comments, deleting the percentages.

Regarding Table 1, since you have removed the values for weight and stature, you no longer need to divide the table in two.

Reviewer 2 Report

Thanks to the authors for the exciting research.

I think the manuscript can be published in the last form.

Sincerely.

Reviewer 3 Report

I sent to you a point by post review rapport. I was wainting from you a point by point answer and this wasn't happening. Anyway, some of my observations were accepted and I congratulated you for that. But, I made some comments and I observed that you didn't correct it. I don't know what to say more that I already did regarding some technical mistakes. 

I am just wondering why you want to keep, in the end of your paperwork, the sentence:

"Citations and references in the Supplementary Materials are permitted provided that they also appear in the reference list here. In the text, reference numbers should be placed in square brackets [] and placed before the punctuation; for example [1], [1–3] or [1,3]. For embedded citations in the text with pagination, use both parentheses and brackets to indicate the reference number and page numbers; for example [5] (p. 10), or [6] (pp. 101–105)."

Also, I am wondering why you don't want to correct the model of references. Some of its are correctly mentioned in references list, and other, don't. 

But, overall, the paperwork was improved. I will ask the editor of the journal to decide about publication in this form. 
